# Physical Activity among Elderly Teachers Working in Basic Education Schools

**DOI:** 10.3390/bs13100841

**Published:** 2023-10-14

**Authors:** Nayra Suze Souza e Silva, Luana Lemos Leão, Rose Elizabeth Cabral Barbosa, Rosângela Ramos Veloso Silva, Tatiana Almeida de Magalhães, Cristina Andrade Sampaio, Luiza Augusta Rosa Rossi-Barbosa, Adriane Mesquita de Medeiros, Desirée Sant’Ana Haikal

**Affiliations:** 1Centre of Biological Sciences and Health, Department of Physical Education, State University of Montes, Claros, Montes Claros 39401-089, MG, Brazil; 2School of Nutrition Sciences, Faculty of Health Sciences, University of Ottawa, Ottawa, ON K1N 6N5, Canada; leaoluanalemos@gmail.com; 3Graduate Program in Health Sciences, Centre of Biological Sciences and Health, State University of Montes Claros, Montes Claros 39401-089, MG, Brazil; rosebarbosa.moc@gmail.com (R.E.C.B.); tatimagmoc@gmail.com (T.A.d.M.); cristina.sampaio@unimontes.br (C.A.S.); desiree.haikal@unimontes.br (D.S.H.); 4Graduate Program in Primary Health Care, Centre of Biological Sciences and Health, State University of Montes Claros, Montes Claros 39401-089, MG, Brazil; rosangela.veloso@unimontes.br (R.R.V.S.); luiza.rossi@funorte.edu.br (L.A.R.R.-B.); 5Graduate Program in Speech-Language Sciences, Federal University of Minas Gerais, Belo Horizonte 31270-901, MG, Brazil; adrianemmedeiros@hotmail.com

**Keywords:** elderly, health of the elderly, physical activity, school teachers

## Abstract

This study aimed to assess the levels of physical activity (PA) among elderly teachers. It was an epidemiological, cross-sectional, and analytical web survey conducted with teachers working in public basic education schools. Data collection took place between October and December 2021 through an online form. The dependent variable was physical activity practice, classifying teachers as either physically active or inactive. Descriptive, bivariate, and multiple analyses were performed using Poisson Regression with robust variance. A total of 1907 teachers participated in the study, of whom 5.6% were elderly, ranging in age from 60 to 72 years. Regarding PA practice, elderly teachers were found to be more physically inactive compared with adult teachers (PR = 1.18; 95% CI = 1.04; 1.34). Conclusion: A statistically significant difference in PA practice was observed between adult and elderly teachers, indicating that elderly teachers are more physically inactive.

## 1. Introduction

Based on research among teachers, high workloads and negative experiences with students and parents are major contributors to teachers’ mental strain [1,2]. Studies have shown a correlation between work-related factors and diminished mental and physical well-being among school teachers [3,4]. Several factors have been linked to a decline in overall well-being, including perceived stress, excessive workload, lack of collegial support, and reduced job satisfaction [5]. Additionally, teachers’ physical well-being is influenced by their gender (female teachers are more susceptible to musculoskeletal pain and poorer mental well-being, for example), the length of their teaching careers, prolonged periods of standing, and a head-down posture, with these factors worsening as they age [6]. Engaging in regular physical activity (PA) has been suggested as a potential solution to address these health and work-related challenges faced by teachers, including recovery from work, to mitigate the relationship between high professional demands and health problems [7,8]. Teachers encounter professional demands that include time pressure, behavioral problems, low student motivation, lack of recognition, and autonomy. Additionally, they dedicate a significant amount of time to work-related activities outside of formal working hours, which limits the time available for work recovery [3,8]. It is known that recovery from work is crucial for extending professional careers because it is closely linked to health and well-being, and aging can slow down the physiological recovery process [8]. In this context, with an increasing number of aging workers in the workforce, it is crucial to understand their challenges and develop strategies to support them and prevent early retirement [8].

As defined by the American College of Sports Medicine, PA is “any movement produced by skeletal muscles that generates energy expenditure”, including exercises, sports, and daily living activities (e.g., active commuting), including worksite PA (i.e., that occurs during working hours and within the workplace) [9]. Numerous systematic reviews have consistently emphasized the importance of regular PA and exercise, especially in in preventing heart disease and various other chronic conditions [10,11,12,13]. More than 25 chronic medical conditions have been identified that may benefit from regular PA and/or exercise [14,15,16,17]. There is a well-established dose–response relationship between PA and health, resulting in consistent reductions in premature mortality and chronic diseases [18,19].

A sedentary lifestyle and insufficient PA rank among the most significant public health issues that need to be addressed to support healthy aging, according to the World Health Organization (WHO) [20]. Research has shown that a sedentary lifestyle can lead to a reduction in maximum aerobic capacity, muscle strength, motor responses, and overall functional capacity [21]. Consequently, the aging process associated with a sedentary lifestyle favors social, mental, and physical dependence [22]. A cross-sectional study using data from the 2019 Brazilian National Health Survey among 22,726 adults aged 60 years or older, of both sexes, showed that 56.0% (95% CI, 55.0–57.1) were physically inactive, while 44% (95% CI, 42.9–45.0) were active [23]. In general, only a small percentage of teachers meet moderate-to-vigorous PA recommendations despite the benefits of PA on physical, social, and psychological health [24,25]. This percentage is even lower among older teachers. A study conducted with teachers in the São Paulo state education network revealed that teachers aged between 55 and 66 years old had lower levels of PA, regardless of the intensity (low, moderate, or high), compared with younger colleagues [24]. Therefore, recognizing the significance of regular PA in the aging process, this study aimed to assess the practice of PA among elderly teachers in the state of Minas Gerais, Brazil.

## 2. Materials and Methods

### 2.1. Study Design

This epidemiological and cross-sectional study followed the guidelines for Reporting Results of Internet E-Surveys [26] and Reporting of Observational Studies in Epidemiology [27].

### 2.2. Participants and Sample Size

The study was conducted with a population of approximately 90,000 active teachers in the public state basic education schools of Minas Gerais, Brazil (data provided by the State Department of Education of Minas Gerais—SEE/MG), distributed across about 3500 schools. The state of Minas Gerais is composed of 853 municipalities, with a population of 20,538,718 inhabitants according to the 2022 census [28]. The ProfSMinas Project was authorized by SEE/MG (State Department of Education of Minas Gerais) and approved by the Research Ethics Committee of the State University of Montes Claros—Unimontes, No. 4,964,125/2021. Along with the data collection form, an Informed Consent Form (ICF) regarding participation in the research was presented. The teachers were also asked whether they accepted or declined to participate in the research (yes or no), as well as given the option to print the ICF, duly signed by the research coordinator, if they wished to do so. To calculate the sample size, a formula based on disease or event prevalence was used, considering an infinite population. A prevalence of 50% was considered with the intention of obtaining the largest sample size and, consequently, greater inferential power for various variables, as the study encompassed multiple outcomes to be investigated. A margin of error of 3% was adopted to reduce error size, thereby increasing the sample size to achieve slightly more precise estimates. Additionally, a 20% increase in the sample size was applied to account for potential losses (non-response rate), which could compromise the validity of the study. In order to ensure that the data collected were representative of Minas Gerais, 1282 teachers needed to be interviewed.

Data collection took place from 26 October to 31 December 2021, carried out online through the Google Forms platform. The study included teachers actively engaged in teaching during the data collection year, working in elementary and/or secondary education, and having an affiliation with one of the state schools in Minas Gerais. As exclusion criteria, the study did not include teachers who were in roles other than teaching (e.g., principals), retirees, and those who declined to participate in the study.

### 2.3. Instruments and Variables

The dependent variable of this study was the practice of PA among teachers, assessed through the International Physical Activity Questionnaire (IPAQ), short version, validated for the Brazilian population [29]. According to the guidelines of WHO for PA, teachers who engaged in at least 150 min of PA per week were classified as active, while those who performed less than 150 min or did not engage in any PA were classified as inactive [30].

The independent variable was the age group (adults—less than 60 years; elderly—60 years or older) [31]. The age group variable is numerical in nature and was categorized in this study according to the Statute of the Elderly for the Brazilian population, which defines the age of the elderly as equal to or greater than 60 years [31].

Adjustment variables: sex (female; male), marital status (married or cohabitant; single; divorced or widowed), diet (inadequate; needs modifications; healthy) [32], oral health (excellent; good; fair; poor; very poor), sleep quality (very good; good; poor; very poor), anxiety (no; yes), depression (no; yes), hypertension (no; yes), heart disease (no; yes), and quality of life (excellent; very good; good; poor; very poor).

Oral health and sleep quality were assessed through the following questions: “How would you rate your oral health?” and “During the last month, how would you rate the overall quality of your sleep?” For the anxiety and depression variables, teachers were asked if they had received a formal diagnosis for these mental health issues in the past 12 months leading up to the data collection for the survey.

The scale used to assess the diet is a validated instrument consisting of 24 questions designed to measure healthy eating practices in accordance with the recommendations of the Brazilian Dietary Guidelines for the population [33]. The questions encompass dimensions of planning, household organization, eating patterns, and food choices. Response options are presented on a four-point Likert scale. The total score is derived from the sum of items (1 to 24), categorized as follows: inadequate diet (up to 31 points), needs modification (between 31 and 41 points), and healthy diet (above 41 points). Exploratory factor analysis (EFA) and confirmatory factor analysis (CFA) were performed to determine construct validity. Internal consistency was determined using α and ω coefficients, and reproducibility was tested using test–retest [32].

### 2.4. Statistical Analysis

All analyses were conducted using the SPSS^®^ 22.0 software. The frequency and prevalence of the variables were presented. Bivariate analyses were conducted using the Pearson chi-square test to observe the relationship between age classification and the other variables used in this study. Poisson Regression with robust variance was employed to associate the levels of PA (with ‘active’ as the reference category) of the teachers with their age classification. Thus, the results were presented as both crude and adjusted prevalence ratios (PR). In the adjusted analysis, the following variables were utilized: sex, marital status, diet, oral health, sleep quality, anxiety, depression, hypertension, heart disease, and quality of life. In addition to the crude and adjusted PR, the 95% confidence interval and a significance level of 5% (α ≤ 0.05) were reported.

To assess the model’s quality, the Deviance test was employed, which evaluates whether the values predicted by the model deviate from the observed values in a manner that Poisson distribution does not predict. Thus, if the *p*-value of the goodness-of-fit test is greater than the adopted significance level (α > 0.05), the null hypothesis that the Poisson distribution provides a good fit can be accepted [34]. The multicollinearity among independent variables was tested, and the results obtained did not indicate correlations among them.

## 3. Results

The study included 1907 teachers from approximately 354 cities in Minas Gerais, with 10.6% working in rural schools. Of the total number of teachers, 5.6% (*n* = 106) were elderly, with an average age of 62.5 ± 2.5 years, ranging from 60 to 72 years.

Considering only the elderly teachers, 67.0% were female, 50.0% were divorced or widowed, 37.7% reported having received a medical diagnosis of anxiety, and 74.0% were physically inactive. The data presented in Table 1 are described along with other information.

The data in Table 2 show the crude and adjusted prevalence ratio of age classification of teachers in relation to PA practice, with adjustment for other variables of interest (as indicated in the table’s footer).

According to the data presented in the aforementioned table, elderly teachers are more physically inactive when compared with adult teachers, aged under 60 years, both in the crude analysis (PR = 1.15; 95% CI: 1.02; 1.30) and the adjusted analysis (PR = 1.18; 95% CI: 1.04; 1.34). The adjusted prevalence ratio of 1.18 resulting from the model indicates that elderly teachers have an 18% higher prevalence of not engaging in physical activity when compared with adult teachers. The Deviance test statistic demonstrated a proper fit for the final multiple model, with a *p*-value greater than 0.05.

## 4. Discussion

The present study aimed to investigate the association between physical activity (PA) practice and the age classification of teachers working in basic education. Consequently, the primary finding of the study is a higher prevalence of physical inactivity among elderly teachers when compared with adult teachers. In line with this finding, a prior meta-analysis reported that only 43% of non-teacher elderly individuals were physically inactive. However, the present study indicates that a striking 74% of elderly teachers are physically inactive [35]. The significant proportion of elderly teachers who do not engage in PA raises concerns about potential adverse impacts on their physical and mental health, as physical inactivity is linked to various health issues, including cardiovascular disease, diabetes, and musculoskeletal disorders [36]. Furthermore, a lack of PA can contribute to cognitive decline and a reduced quality of life, which can affect teaching effectiveness and interactions with students [37]. Maintaining PA levels during aging is crucial for both physical and mental health, as PA practices enable the prevention, reversal, and mitigation of the typical problems associated with aging [38].

Considering health issues, the elderly teachers in this study exhibited a higher prevalence of hypertension and heart disease, and physical inactivity was found to be associated with these and other diseases, such as obesity and diabetes [39]. The likelihood of developing hypertension increases with advancing age, with a prevalence of over 60% in developed countries, making hypertension the most common chronic disease among the elderly [40]. Additionally, there is an observable association between hypertension and cardiovascular risk factors in the elderly. The Framingham study highlighted a higher percentage of hypertensive patients with other cardiac problems (30%) when compared with those with hypertension alone (19%) [41]. Physical inactivity serves as a significant risk factor for diseases related to decreased bodily function. Therefore, promoting the adoption of regular PA among the elderly would not only result in improved longevity but also greater energy and the ability to perform daily activities [42,43].

Being a teacher in the present day entails complex and significant responsibilities, as educators must prepare children and youth for a constantly changing and evolving society [44]. Teachers play a pivotal role in the educational process, underscoring the importance of research into their well-being, particularly their mental health [45]. In our study, 37.7% of elderly teachers reported receiving an anxiety diagnosis. Many aspects of the educational environment can be challenging and stressful, with teachers often facing pressure to meet teaching goals, deal with student and administrative demands, and deal with other professional difficulties [46]. These factors may contribute to anxiety among elderly teachers. Work-related stress, the increasing need for innovation and adaptation to changes in the educational system, and the responsibilities of overseeing numerous and diverse classes may all contribute to anxiety among elderly teachers [47].

The presence of elderly teachers, however, can also pose challenges as well, such as adapting to modern technological and pedagogical advancements [48]. Ensuring that these professionals have access to ongoing development opportunities is crucial for them to continue providing quality education.

Additionally, it is worth noting that 50% of the elderly teachers evaluated in this study were either divorced or widowed. Divorce or widowhood can have significant implications for the emotional well-being of elderly teachers [49]. Losing a partner or going through a divorce can increase feelings of stress, anxiety, and loneliness [50,51]. These factors may potentially have adverse effects on teachers’ mental health and their ability to effectively perform their educational duties [5]. Furthermore, marital situations can also influence an individual’s level of PA, with women often being physically active due to household chores and men engaging in PA through occupational and external domestic tasks like gardening, yard cleaning, car washing, etc. [52].

In the United States, the CDC’s Prevention Research Centers-Healthy Aging Research Network has developed community-based PA programs for older adults, aiming to maintain or improve functional abilities and help the elderly lead independent lives. Certified, trained fitness instructors lead the classes, which are held three times a week on an ongoing basis [53]. In Brazil, the Health Academy Program is one of the health care facilities within primary health care, which serves as the gateway for users to the Brazilian Health System (Sistema Único de Saúde [SUS]) [54]. The program’s centers consist of physical structures, equipment, and qualified professionals dedicated to providing services related to health diagnosis, treatment, prevention, and promotion for the population, including the elderly. Furthermore, in light of the findings of this study, there is a compelling case for implementing PA incentivization and intervention programs tailored to the needs of elderly teachers. These programs could serve as proactive measures to promote PA and improve the overall health and well-being of this demographic.

Some limitations need to be considered in this study. Since it is a web survey study, it is prudent to consider potential selection and memory biases, as the research relies on Internet access to complete the survey, and the responses are self-reported. However, online surveys also offer advantages, such as broader geographic coverage and faster dissemination of results. Additionally, the fact that the sample consists of teachers working in schools in only one Brazilian state presents a limitation, as the demographic, socioeconomic, and cultural characteristics of that state may differ from those of other regions, potentially affecting the generalizability of the results to different contexts.

## 5. Conclusions

This study addressed the practice of PA among teachers, which revealed a substantial difference in PA levels between elderly and younger teachers. In both crude and adjusted analyses, elderly teachers were found to be more physically inactive, even after considering various relevant variables. Furthermore, it was found that in addition to being more physically inactive, when compared with adult teachers, elderly teachers also had a higher prevalence of hypertension and heart disease.

Nevertheless, this study highlights an opportunity for positive change, ensuring that our aging teacher population can continue to thrive in their vital roles within the education system. Developing PA programs customized to the requirements of elderly teachers, along with increasing awareness about the advantages of exercise, may be effective strategies for encouraging PA among older educators. These interventions may not only improve the health of older teachers but may also contribute to a healthier and more productive teaching environment.

## Figures and Tables

**Table 1 behavsci-13-00841-t001:** Bivariate analysis according to age classification of basic education teachers. Minas Gerais, 2021 (*n* = 1907).

Variables		Age Group *		
		Adults	Elderly	
	*n* (%)	*n* (%)	*n* (%)	*p*-Value
Sex				0.008
Female	1473 (77.2)	1401 (77.8)	71 (67.0)	
Male	434 (22.8)	399 (22.2)	33 (33.0)	
Marital status				<0.001
Married or cohabitant	276 (14.5)	238 (13.2)	38 (35.8)	
Single	471 (24.7)	456 (25.3)	15 (14.2)	
Divorced or widowed	1160 (60.8)	1106 (61.4)	53 (50.0)	
Diet				0.251
Inadequate	198 (10.4)	191 (10.6)	7 (6.6)	
Needs modifications	738 (38.7)	700 (38.9)	38 (35.8)	
Healthy	971 (50.9)	909 (50.5)	61 (57.5)	
Oral health				0.485
Excellent	307 (16.1)	289 (16.1)	17 (16.0)	
Good	1008 (52.9)	952 (52.9)	56 (52.8)	
Regular	483 (25.3)	460 (25.6)	23 (21.7)	
Poor	91 (4.8)	83 (4.6)	8 (7.5)	
Very poor	18 (0.9)	16 (0.9)	2 (1.9)	
Sleep quality				0.201
Very good	281 (14.7)	260 (14.4)	21 (19.8)	
Good	879 (46.1)	826 (45.9)	52 (49.1)	
Poor	564 (29.6)	537 (29.8)	27 (25.5)	
Very poor	183 (9.6)	177 (9.8)	6 (5.7)	
Anxiety				0.467
No	1171 (61.4)	1104 (61.3)	66 (62.3)	
Yes	736 (38.6)	696 (38.7)	40 (37.7)	
Depression				0.411
No	1658 (86.9)	1566 (87.0)	91 (85.8)	
Yes	249 (13.1)	234 (13.0)	15 (14.2)	
Hypertension				<0.001
No	1483 (77.8)	1428 (79.3)	55 (51.9)	
Yes	424 (22.2)	372 (20.7)	51 (48.1)	
Heart disease				<0.001
No	1830 (96.0)	1736 (96.4)	94 (88.7)	
Yes	77 (4.0)	64 (3.6)	12 (11.3)	
Quality of life				0.401
Excellent	137 (7.2)	126 (7.0)	11 (10.4)	
Very good	594 (31.1)	555 (30.8)	38 (35.8)	
Good	1073 (56.3)	1021 (56.7)	52 (49.1)	
Poor	93 (4.9)	88 (4.9)	5 (4.7)	
Very poor	10 (0.5)	10 (0.6)	0 (0.0)	
Physical activity level *				0.031
Active	625 (35.0)	599 (35.5)	26 (26.0)	
Inactive	1162 (65.0)	1087 (64.5)	74 (74.0)	

* Variation in *n* due to loss of information.

**Table 2 behavsci-13-00841-t002:** Prevalence ratio between adult and elderly teachers according to physical inactivity. Minas Gerais, 2021 (*n* = 1787).

Variable	PR_unadjusted_ (CI95%)	*p*-Value	PR_adjusted_ (CI95%)	*p*-Value
Age group *		0.026		0.011
Adults	1.00		1.00	
Elderly	1.15 (1.02; 1.30)		1.18 (1.04; 1.34)	
Deviance: 0.547/*p*-value: 0.352				

PR, prevalence ratio; 95% CI, 95% confidence interval. The analysis was adjusted for sex, marital status, diet, oral health, sleep quality, anxiety, depression, hypertension, heart disease, and quality of life. * Variation in *n* due to loss of information.

## Data Availability

The data presented in this study are available from the corresponding author upon reasonable request.

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
