# Peer review of "Physical Activity among Elderly Teachers Working in Basic Education Schools"

_behavsci, 2023, doi:10.3390/bs13100841_

Round 1

Reviewer 1 Report

The authors investigated the PA levels among elderly teachers and found that elderly teachers were more physically inactive than adult teachers, and hence, more physical problems may occur. This study is quite interesting; however, I think there are some critical issues that the authors need to address before it reaches publication. Here are my concerns:

1.      Page 1 Introduction, the author mentioned, “In this sense, with more aging workers in the workforce, it is crucial to understand their challenges and to develop strategies to support them and prevent early retirement [8].”. The rationale to link the preview to elderly teachers is not sufficient, please add more to justify why it is vital to understand the challenges, are these the assumptions that the elderly teachers are also facing similar situations did the others?

2.      Page 2 Introduction, the authors claimed that “only a small percentage of teachers meet moderate…”, It would be better to also include the older adults’ pattern in PA in Brazil and to come to the aim of the study by taking these together.

3.      Page 2, 2.2. Participants and sample size, all teachers participating in this study signed an Informed Consent Form. May I know if it is a physical copy or an online version?

4.      Page 2, 2.2. Participants and sample size, please justify why 50% of the prevalence rate is taken and why the margin of error is taken as 3% instead of 5%?

5.      Page 2, the period of data collection should be stated.

6.      Page 2, Instruments and variables, the authors should discuss how to differentiate the age group by adults and elderly.

7.      Page 3, concerning the diet scales, will there be any reliability statistics for the scales?

8.      Page 3, statistical analysis, the authors need to explain why some non-dichotomous variables are transformed into dichotomous ones during the analyses. Moreover, even if the authors do it for convenience, it is more appropriate to show all the details of each variable, for example, how many people fall into excellent oral health, good oral health, and so on.

9.      Page 3, statistical analysis, the authors used PR to investigate the magnitude of association, however, in Table 2 in the results section, the authors did not explain how these variables were controlled or adjusted, is that a hierarchical logistic regression or what?  

10.  Page 3, Results, please explain why the cut-off for older adults is 60 instead of 65.

11.  Page 4, Table 1, for the dependent variable, PA level, it was indicated that there were missing data, however, for Table 2, how did the N size become 1,907 if participants were not reporting that question?

12.  Page 4, discussion, please explain more in detail what PR=1.18 indicate.

13.  Page 5, discussion, I personally think that the discussion about the gender difference in engagement in teaching activities is irrelevant to the topic concerned.

14.  Page 5, discussion, the authors mentioned suggestions for launching PA programs in conclusion while I think it is more crucial to be discussed in the discussion part, the authors should address some successful programs in this section as well.

Some minor grammatical mistakes were observed. 

Author Response

We appreciate your feedback and are committed to improve the quality of our research. Your input is valuable to us; therefore, your suggestions were answered bellow.

Page 1 Introduction, the author mentioned, “In this sense, with more aging workers in the workforce, it is crucial to understand their challenges and to develop strategies to support them and prevent early retirement [8].”. The rationale to link the preview to elderly teachers is not sufficient, please add more to justify why it is vital to understand the challenges, are these the assumptions that the elderly teachers are also facing similar situations did the others?

More information about the challenges and their significance has been added to the text.

Page 2 Introduction, the authors claimed that “only a small percentage of teachers meet moderate…”, It would be better to also include the older adults’ pattern in PA in Brazil and to come to the aim of the study by taking these together.

The information has been inserted into the text.

Page 2, 2.2. Participants and sample size, all teachers participating in this study signed an Informed Consent Form. May I know if it is a physical copy or an online version?

The procedures were all conducted online. Teachers received, along with the data collection instrument, a copy of the Consent Form signed by the responsible researcher, available for download, and all those who indicated their acceptance to participate in the study in the data collection instrument were included. The information has been inserted into the text.

Page 2, 2.2. Participants and sample size, please justify why 50% of the prevalence rate is taken and why the margin of error is taken as 3% instead of 5%?          

A prevalence of 50% was considered with the intention of obtaining the largest sample size and, consequently, greater inferential power for various variables, given that the study encompasses multiple outcomes to be investigated (an umbrella study). Since it is a web survey, a 3% error margin was adopted to reduce error size, increasing the sample size to achieve slightly more precise estimates. Additionally, a 20% increase in the sample size was applied to account for potential losses (non-response rate) that could compromise the validity of the study.

Page 2, the period of data collection should be stated.        

The information has been inserted into the text.

Page 2, Instruments and variables, the authors should discuss how to differentiate the age group by adults and elderly.      

The information has been inserted into the text.

Page 3, concerning the diet scales, will there be any reliability statistics for the scales?           

The information has been inserted into the text.

Page 3, statistical analysis, the authors need to explain why some non-dichotomous variables are transformed into dichotomous ones during the analyses. Moreover, even if the authors do it for convenience, it is more appropriate to show all the details of each variable, for example, how many people fall into excellent oral health, good oral health, and so on.      

The variables that were recategorized have been presented in their original question format.

Page 3, statistical analysis, the authors used PR to investigate the magnitude of association, however, in Table 2 in the results section, the authors did not explain how these variables were controlled or adjusted, is that a hierarchical logistic regression or what? 

Indeed, it is not a hierarchical regression. The aim was to relate physical activity practice to the age group. In this case, to avoid an overly simplistic analysis, we chose to present an adjusted analysis by variables of interest as well. These observations have also been inserted into the text for further clarification.

Page 3, Results, please explain why the cut-off for older adults is 60 instead of 65.           

The information has been included in the text, and we opted for referencing the Brazilian Law, which considers the elderly as equal to or greater than 60 years.

Brasil. Lei nº 10.741, de 1º de outubro de 2003. Dispõe sobre o Estatuto da Pessoa Idosa. Disponível em: https://www.planalto.gov.br/ccivil_03/leis/2003/l10.741.htm.

Page 4, Table 1, for the dependent variable, PA level, it was indicated that there were missing data, however, for Table 2, how did the N size become 1,907 if participants were not reporting that question?     

The information has been inserted into the text.

Page 4, discussion, please explain more in detail what PR=1.18 indicate. 

The information has been inserted into the text.

Page 5, discussion, I personally think that the discussion about the gender difference in engagement in teaching activities is irrelevant to the topic concerned.      

The mentioned paragraph has been removed.

Page 5, discussion, the authors mentioned suggestions for launching PA programs in conclusion while I think it is more crucial to be discussed in the discussion part, the authors should address some successful programs in this section as well. 

A new paragraph was added to the discussion to address this topic.

Comments on the Quality of English Language. Some minor grammatical mistakes were observed.

Dear reviewer, we conducted a thorough proofreading and editing process to correct any mistakes and enhance the overall readability of the manuscript.

Reviewer 2 Report

My comments are given on the file.

Author Response

We appreciate your feedback and are committed to improve the quality of our research. Your input is valuable to us; therefore, your suggestions were answered bellow.

The introduction is very short. Check if there are studies in similar fields in the bibliography that confirm your objectives or hypotheses.  

The information has been inserted into the text.

The study could compare the elderly professors’ FA and elderly non-professor. I think is more interest compare Physical Activity Among Elderly Teachers with the physical activity among elderly population. If there are differences between them?

We appreciate your suggestion, and we concur that comparing the PA levels of elderly teachers with those of the general elderly population would indeed be of interest. However, we lack data for the general (non-teachers) population because our research group's focus is exclusively on teachers. Consequently, the data we have is derived solely from this specific population.

To address the desire for comparison between the prevalence of physical inactivity among elderly teachers and the broader elderly population, we have incorporated information from a recently published meta-analysis in the discussion section.

The statistical methods used are basic (mean differences).

The statistical analysis of the study (Poisson Regression) was more detailed, and two new tests, Deviance and correlation, were also included.

The study’s conclusions are overly concise. Could you please elaborate further and provide additional conclusions?       

The information has been inserted into the text.

Finally, I miss the limitations of the study.  

The study's limitations have been included in the last paragraph of the discussion.

Reviewer 3 Report

Dear authors.

This paper was very well written and has important implications for older teachers and their adherence to WHO physical activity guidelines.

I have attached the paper and suggested one additional reference. Other than that, your paper offers a valuable and insightful contribution to understanding the physical activity patterns among elderly educators. The study employs a well-structured methodology, combining epidemiological, cross-sectional, and analytical approaches to gather data from a substantial sample of teachers working in public basic education schools.

Well done!

Author Response

We appreciate your feedback and are committed to improve the quality of our research. Your input is valuable to us; therefore, your suggestions were answered bellow.

An additional and recent reference that is applicable and takes into account COVID.

Rahayu, Tutiek & Pertiwi, Kartika Ratna & Kushartanti, Wara & Arovah, Novita. (2023). Physical Activity and Post-COVID-19 Syndrome in Older Adults: A Systematic Review. International Journal of Kinesiology and Sports Science. 11. 42-52. 10.7575/aiac.ijkss.v.11n.1p.42. 

The reference has been included.

Round 2

Reviewer 1 Report

The authors addressed all my points. Good luck. 

Reviewer 2 Report

Los autores have incorporated into the article's text all the comments made by the reviewers. If they haven't done so, they have provided a proper explanation for why they deemed it unnecessary. Therefore, my decision is to accept the article.